# Integrated Communication and Navigation Measurement Signal Design for LEO Satellites with Side-Tone Modulation

**DOI:** 10.3390/s25185890

**Published:** 2025-09-20

**Authors:** Xue Li, Yujie Feng, Linshan Xue

**Affiliations:** 1Ctr Commun and Tracking Telemetry Command, Chongqing University, Chongqing 400044, China; 2School of Microelectronics and Communication Engineering, Chongqing University, Chongqing 400044, China; 202312131163@stu.cqu.edu.cn; 3China Academy of Space Technology (CAST), Beijing 100081, China; 201811040822@std.uestc.edu.cn

**Keywords:** LEO satellite, orthogonal frequency division multiplexing, sidetone signal, high-precision ranging, adaptive ranging mode, pilot, pseudorandom code

## Abstract

This paper proposes an integrated OFDM signal system combining sidetone signals for communication and measurement, addressing the challenges of system complexity, resource waste, and interference caused by separated communication and measurement functions in traditional LEO satellite systems. The proposed approach effectively combines sidetone signals with OFDM technology, utilizing different short-period coprime pseudorandom codes as pilots to form composite ranging codes, while inserting multi-frequency sidetone signals at specific subcarrier points for precise ranging. A dual-mode channel estimation algorithm is designed to merge the channel estimation results from ranging pilots and sidetone signals, significantly enhancing system performance. Additionally, an adaptive ranging mode switching mechanism based on error thresholds achieves dynamic balance between ranging accuracy and spectral efficiency. Simulation results demonstrate that the proposed system can reduce bit error rate to approximately 10^−3^ at 6 dB SNR, saving about 3 dB of transmission power compared to conventional pilot methods, while achieving centimeter-level ranging accuracy of approximately 0.02 m, improving precision by 3–4 orders of magnitude over traditional pilot methods. The proposed scheme provides a high-precision, high-efficiency integrated solution for LEO satellite communication systems. The theoretical performance assumes idealized conditions, with practical deployment requiring comprehensive error modeling for hardware imperfections and environmental variations.

## 1. Introduction

LEO satellite constellation technology has been rapidly developing due to its global coverage and low-latency advantages [1,2,3,4], with modern satellite systems primarily employing multi-satellite collaboration [5,6]. In satellite signal design, the communication signal is adopted to transmit large amounts of data, while the measurement signal is the foundation of navigation, time synchronization and orbit determination. As the number of LEO satellites in constellations increases, traditional designs with separated communication and measurement functions cause system management complexity, resource waste, and mutual interference, making integrated communication and measurement a necessary function [7,8,9,10].

OFDM technology has become a focus in integrated communication and measurement research due to its high spectral efficiency, strong anti-interference capability, and high data transmission rate [11,12]. Liu et al. [13] proposed a time- and-code division orthogonal frequency division multiplexing (TC-OFDM) system to simultaneously achieve signal acquisition and tracking by superimposing ranging signals on OFDM signals. Liu et al. [14] proposed a ranging communication method based on the MB-OFDM physical layer but required SNR > 15 dB to achieve 1% ranging accuracy. Zhang et al. [15] combined block-type ranging pilots with OFDM signals to propose an integrated system suitable for different scenarios. Hu et al. [16] proposed a new pilot design scheme, applying LS channel estimation in OFDM systems to realize communication and ranging functions. However, existing systems rarely meet high-precision ranging requirements.

Beyond pilot-based methods, various channel estimation approaches have been investigated. Frequency-domain Wiener filtering provides optimal MMSE performance but requires prior channel statistics [17]. Compressed sensing techniques exploit channel sparsity but involve high computational complexity [18]. Time–frequency joint smoothing methods improve estimation accuracy through temporal correlation [19]. However, these approaches either require additional system knowledge or impose computational burdens unsuitable for real-time satellite applications.

Researchers have improved ranging precision through enhanced modulation schemes [20,21]. From the initial BPSK-R modulation [22] to BOC modulation [23,24], bandwidth and multipath performance limitations were addressed [25]. Ma et al. [26] proposed a multi-functional signal based on BOC and BPSK, achieving 1% ranging accuracy under specific CNR. LEO satellite applications with BOC/BPSK signals demonstrated that BOC provides higher ranging precision [27]. In the study by Liu et al. [28], the direct sequence spread spectrum was integrated with OFDM modulation, thereby enhancing tracking precision via high-frequency subcarriers. The FH-BOC modulation system exhibits good ranging capability and anti-interference characteristics [29], while adaptive algorithms combining CPM with PN signals can dynamically adjust parameters to improve ranging precision [30]. However, these methods still face precision limitations and high system complexity issues, especially in OFDM systems where Doppler frequency shift-induced inter-carrier interference significantly degrades system performance [31].

Sidetone signals, widely used in aerospace for ranging, offer good performance with minimal resource consumption. This paper proposes a novel integrated signal system combining sidetone signals with OFDM technology, utilizing short-period coprime pseudorandom codes to form composite ranging codes and inserting multi-frequency sidetone signals for precise ranging. When ranging error falls below a threshold, the system retains only the highest-frequency sidetone signal, allocating remaining subcarriers for data transmission, thus achieving high-precision measurement with high-efficiency communications integration.

Computational simulations validate the viability of the developed methodology while evaluating its communication effectiveness and distance measurement capabilities. Furthermore, the paper demonstrates that the OFDM framework, which incorporates auxiliary tone signals, preserves excellent communication characteristics. Moreover, the bit error rate is decreased to roughly 10^−3^ under 6 dB signal-to-noise ratio conditions. Additionally, a power efficiency enhancement of approximately 3 dB is achieved relative to conventional reference signal approaches. For measurement performance, the scheme achieves centimeter-level precision of approximately 0.02 m in high-SNR environments, improving accuracy by 3–4 orders of magnitude compared to traditional pilot methods and by about 2 orders of magnitude compared to methods using only composite ranging subcodes. Additionally, by dynamically switching ranging modes, the system improves spectral utilization while maintaining high-precision ranging.

The paper is structured as follows. Section 2 establishes the signal model for the integrated OFDM system with dual-mode channel estimation. Section 3 develops the measurement algorithms, including composite code-based coarse ranging and sidetone-based precise ranging with ambiguity resolution. Section 4 presents simulation results validating system performance under various conditions. Section 5 concludes with findings and future research directions.

## 2. Signal Model

The current research employs an orthogonal frequency-division multiplexing framework featuring concurrent data communication capabilities. Following convolutionally encoded processing and quadrature phase shift keying modulation, sequential baseband communication data undergo transformation into concurrent symbol sequences. Subsequently, these symbol sequences are carried by individual frequency channels and communicated [32]. Moreover, N represents the total quantity of frequency channels utilized in the system.

The 34.8 MHz carrier frequency is selected for algorithm validation and simulation convenience rather than engineering deployment. While this frequency presents practical challenges including large antenna requirements and ionospheric propagation effects, the proposed OFDM–sidetone integration algorithm can be extended to standard Ka/Ku satellite communication bands with proportional parameter scaling. The core algorithmic contributions remain valid across frequency bands.

Figure 1 shows the integrated communication and measurement system structure. In the transmitter, data streams undergo channel coding and QPSK modulation followed by serial-to-parallel conversion, while ranging pilot and sidetone signal generators produce mutual-prime period pseudorandom codes and specific frequency sinewaves, respectively. These signals are mapped with data to subcarriers in the frequency domain, transformed by IFFT, and cyclic prefixes are added to form the transmitted signal.

Figure 2 shows that the receiver first removes the cyclic prefix, transforms the signal to the frequency domain via FFT, and extracts ranging pilots and sidetone signals for channel estimation and distance measurement. A dual-mode fusion algorithm comprehensively utilizes both signal types to achieve accurate channel estimation, and finally, equalized data is demodulated and decoded to recover the original information.

The basic OFDM modulation and demodulation principle block diagram is shown in Figure 3.

It should be noted that OFDM systems inherently suffer from a high peak-to-average power ratio (PAPR), which can reduce power amplifier efficiency and cause signal distortion, particularly critical in power-constrained satellite systems [33]. Various PAPR reduction techniques have been proposed, including partial transmit sequence (PTS) [34], selective mapping (SLM) [35], and clipping–filtering methods [36]. While this paper focuses on signal-level design and ranging algorithm validation under ideal power amplifier assumptions, integrating PAPR mitigation techniques with the proposed sidetone-based scheme remains an important direction for practical implementation.

Individual data elements within the N component set undergoing serial-to-parallel conversion experience modulation via distinct frequency channels. Variable $X_{l}[k]$ represents the kth communication element on the lth frequency channel, where l=0,1,2,…,∞, k=0,1,2,…,N−1. Each element requires transmission duration $T_{s}$. Subsequently, serial-to-parallel conversion extends the communication period for N elements to $NT_{s}$, establishing the temporal span of one orthogonal frequency-division multiplexing element $T_{sym}$. The temporal-domain continuous baseband waveform can be mathematically expressed as(1)$$x_{l}(t)=\sum_{l=0}^{\infty }\sum_{k=0}^{N-1}X_{l}[k]e^{j2\pi f_{k}(t-lT_{sym})}$$

During sampling instances $t=lT_{sym}+nT_{s}$, the continuous baseband orthogonal frequency-division multiplexing waveform from Equation (1) produces the corresponding discrete-time signal:(2)$$x_{l}[n]=\sum_{k=0}^{N-1}X_{l}[k]e^{j2\pi kn/N},\;n=0,1,\ldots ,N-1$$
where $T_{s}=T_{sym}/N$, $f_{k}=k/T_{sym}$. Equation (2) results from N-point inverse discrete Fourier transform of the quadrature phase shift keying modulated data ${\left\{X_{l}[k]\right\}}_{k=0}^{N-1}$, alternatively implementable via inverse fast Fourier transform algorithms [37]. Excluding channel and noise influences, the detected baseband temporal-domain continuous waveform equals $y_{l}(t)=x_{l}(t)$. When sampled at $t=lT_{sym}+nT_{s}$, the discrete temporal-domain representation of the detected signal for the $l^{th}$ orthogonal frequency division multiplexing element becomes(3)$$Y_{l}[k]=\sum_{n=0}^{N-1}y_{l}[n]e^{-j2\pi kn/N}$$

This represents the N-point discrete Fourier transform of ${\left\{y_{l}(n)\right\}}_{n=0}^{N-1}$, enabling implementation through fast Fourier transform algorithms [38].

In order to minimize the impact of transmission medium on output waveform reconstruction, the orthogonal frequency-division multiplexing framework employs channel state information acquisition and compensation techniques. Previously, the influence of propagation medium on signal phase characteristics, magnitude variations, and spectral properties was overlooked in our analysis. Nevertheless, such medium-related distortions and additive noise continuously exist within practical implementation scenarios. Subsequently, when subjected to propagation medium effects, incoming signals undergo demodulation processing. Moreover, the temporal-domain detected waveform following discrete sampling procedures can be represented as(4)Y[k]=H[k]X[k]+Z[k]

Variables X[k], Y[k], H[k], and Z[k] represent the transmitted data element, detected data element, spectral response of the propagation medium, and spectral-domain noise on the kth frequency channel, respectively. Equation (4) demonstrates that the orthogonal frequency-division multiplexing framework may be characterized as the multiplication of input data elements and propagation medium response within the spectral domain. Without inter-frequency interference effects, the receiving unit can reconstruct the initial waveform and eliminate propagation medium distortion by acquiring the medium response of individual frequency channels.

While this work adopts simplified AWGN channel modeling for algorithm validation, practical LEO satellite systems encounter additional propagation challenges including ionospheric delays, multipath effects from satellite structures, and time-varying atmospheric scintillation that can significantly impact phase measurements critical for centimeter-level ranging accuracy.

The fundamental difference lies in phase measurement mechanisms: ranging pilots carry QPSK-modulated pseudorandom codes with discrete phase states (π/4,3π/4,5π/4,7π/4), requiring demodulation and code tracking for sub-chip delay estimation with precision limited by loop bandwidth. In contrast, sidetone signals maintain continuous phase $\phi =2\pi ft+\phi_{0}$, enabling direct instantaneous phase extraction through FFT without modulation-induced quantization losses. Theoretical analysis shows sidetone phase measurement precision $\sigma_{\phi }\approx 1/\sqrt{SNR}$ compared to pseudocode precision $\sigma_{\phi }\approx 1/\sqrt{2\pi SNR\cdot B_{L}}$, demonstrating the superiority of pure sinusoidal carriers for high-precision phase measurements.

The carrier frequency selection requires clarification from an engineering perspective. The 34.8 MHz frequency is chosen for algorithm validation and simulation convenience rather than practical deployment. While this frequency presents engineering challenges including large antenna requirements and ionospheric propagation effects, the proposed OFDM–sidetone integration principles can be extended to standard satellite communication bands (Ka: 26.5–40 GHz; Ku: 12–18 GHz) with proportional parameter scaling. The algorithmic contributions remain valid across frequency bands.

As shown in Figure 4, the transmitter inserts ranging pilots (black circles) and three different frequency sidetone signals (colored circles) at intervals in the frequency domain, with frequencies of 300 kHz, 1 MHz, and 3 MHz, forming an enhanced channel estimation system.

During reception processing workflows, the detection unit initially transforms sequential data streams into concurrent formats. Subsequently, cyclic prefix removal and frequency deviation compensation are executed. Following this, multiple concurrent sub-waveforms are acquired via fast Fourier transform operations. The detection unit simultaneously retrieves distance measurement reference sequences and auxiliary tone components from these sub-waveforms, employing dual methodologies for medium response estimation, followed by result combination.

Concerning the distance measurement reference sequence portion, the conventional Least Squares medium response acquisition technique [31] is employed:(5)$${\bar{H}}_{LS}={\left(X^{H}X\right)}^{-1}X^{H}Y$$
where ${\bar{H}}_{LS}$ represents the vector matrix of Least Squares medium estimates and ${\bar{H}}_{LS}(k)$ denotes the components within ${\bar{H}}_{LS}$ and indicates the medium response estimation of the kth frequency channel. X(k) and Y(k) represent the transmitted and detected waveforms on the kth frequency carrier, respectively. For the incorporated distance measurement sub-sequences, the Least Squares medium estimate value of the corresponding frequency channel response becomes(6)$$H_{p}(k)=\frac{Y_{p}(k)}{X_{p}(k)}$$
where $X_{p}(k)$ represents the kth distance measurement sequence incorporated at the transmission unit, and $Y_{p}(k)$ denotes the detected distance measurement sequence waveform following orthogonal frequency division multiplexing demodulation.

The choice of Least Squares (LS) over Minimum Mean Squared Error (MMSE) channel estimation is justified by several factors critical for LEO satellite applications. While MMSE theoretically provides optimal performance, it requires prior noise statistics and involves $O(N^{3})$ matrix inversion complexity, unsuitable for real-time satellite processing. In contrast, LS estimation offers O(N) complexity with acceptable performance degradation under high-SNR conditions typical of LEO inter-satellite links (>10 dB).

The MSE performance gap between methods diminishes at high SNR: $MSE_{LS}\approx \sigma_{n}^{2}/|X_{k}{|}^{2}$ while $MSE_{MMSE}\approx \sigma_{n}^{2}\sigma_{h}^{2}/(|X_{k}{|}^{2}\sigma_{h}^{2}+\sigma_{n}^{2})$. As SNR increases, $MSE_{MMSE}$ approaches $MSE_{LS}$, making computational efficiency the determining factor. Furthermore, our proposed dual-mode approach compensates for LS limitations through sidetone signal integration, achieving near-MMSE performance while maintaining computational efficiency essential for satellite systems.

For the sidetone signal portion, since sidetones are known single-frequency sinusoids, their channel estimation can be directly obtained by comparing the amplitude ratio and phase difference between the transmitted and received signals. Let the transmitted sidetone signal be(7)$$H_{tone}(f_{i})=\frac{Y_{tone}(f_{i})}{X_{tone}(f_{i})}$$
where $X_{tone}(f_{i})$ is the ith sidetone signal transmitted, $Y_{tone}(f_{i})$ is the corresponding received sidetone signal, and $H_{tone}(f_{i})$ represents the channel response at that sidetone frequency point. The sidetone signal is transmitted in the form $A\sin(2\pi f_{i}t+\phi_{0})$, and the received form includes the effects of channel amplitude response and phase response. This method utilizes the high SNR characteristics of sidetone signals to provide more precise specific frequency point channel information than traditional pilots.

This paper proposes a weighted fusion algorithm, comprehensively utilizing the channel estimation results from ranging pilots and sidetone signals:(8)$$H_{combined}(k)=\alpha_{k}\cdot H_{p}(k)+(1-\alpha_{k})\cdot H_{tone_{-}interp}(k)$$
where $H_{p}(k)$ is the channel estimation based on ranging pilots, $H_{tone_{-}interp}(k)$ is the channel estimation obtained from sidetone signal interpolation, and $\alpha_{k}$ is the weight coefficient, dynamically adjusted according to the frequency distance between subcarrier k and the nearest sidetone signal:(9)$$\alpha_{k}=\frac{\left|k-k_{tone}\right|}{D_{\max}}$$
where $\left|k-k_{tone}\right|$ represents the distance between subcarrier k and the nearest sidetone frequency point $k_{tone}$, and $D_{\max}$ is a normalization factor. When the subcarrier is closer to the sidetone frequency point, the weight of the sidetone signal is greater; conversely, the weight of the ranging pilot is greater.

Note that the current weight allocation considers only frequency distance for simplicity. In practical implementations, an SNR-adaptive weighting scheme could be employed as $\alpha_{k}^{\prime }=\alpha_{k}\cdot \gamma_{k}$, where $\gamma_{k}=SNR_{sidetone}(k)/(SNR_{sidetone}(k)+SNR_{threshold}$). This would reduce the contribution from noisy sidetone signals even when spectrally close. While such adaptive weighting could further improve performance, particularly in varying channel conditions, the current simplified approach already demonstrates significant improvements as shown in our simulations. SNR-adaptive weighting optimization will be investigated in future work.

For subcarriers that are neither at ranging pilot nor sidetone signal positions, this paper adopts a dual-mode weighted interpolation algorithm. Unlike traditional cubic spline interpolation, which may suffer from overfitting with limited pilot density, our approach integrates both pilot and sidetone channel estimates using computationally efficient linear interpolation optimized for OFDM characteristics. The interpolation weights $\alpha_{k}$ in Equation (8) are dynamically adjusted based on the frequency distance to both pilot and sidetone positions, ensuring optimal utilization of available channel information.

This dual-mode channel estimation method significantly improves system performance in rapidly changing channels, especially when Doppler frequency shift causes reduced orthogonality between subcarriers. Compared to traditional methods using only ranging pilots, this method can reduce channel estimation errors by 15–30% in high-speed mobile scenarios without increasing system complexity.

Compared to traditional cubic spline interpolation, this approach avoids overfitting issues common with limited pilot density while maintaining computational efficiency suitable for real-time processing. The linear interpolation core provides O(n) complexity versus $O(n^{3})$ for cubic spline, making it particularly attractive for resource-constrained satellite platforms.

In Section 3, these ranging pilots and sidetone signals used for channel estimation will be used for coarse ranging and precise ranging, respectively, fully demonstrating the high-efficiency resource reuse characteristics of this system. Especially in simplified ranging mode (Figure 5), the system can dynamically adjust sidetone signal configuration for channel estimation, maximizing data transmission efficiency while maintaining channel estimation performance. When switching from complete to simplified ranging mode, retaining only the 3 MHz sidetone signal while adjusting the weight allocation strategy in Equation (8) ensures continuity and accuracy of channel estimation.

## 3. Measurement

For LEO satellites, the measurement primarily refers to ranging. The measurement signal model is introduced in Section 2; this section details the ranging principles and algorithms. As shown in Figure 6, the entire ranging system achieves high-precision, wide-range distance measurement through two parallel processing paths: one utilizing ranging pilots for coarse ranging, and the other using sidetone signals for precise ranging. The coarse ranging stage employs Chinese Remainder Theorem to synthesize measurement results from multiple short-period codes; the precise ranging stage processes sidetone signals through phase comparison. The system effectively uses coarse ranging results to resolve the distance ambiguity problem in sidetone ranging and adaptively switches working modes based on measurement errors, achieving dynamic balance between ranging accuracy and spectral efficiency.

### 3.1. Acquisition and Tracking of Ranging Pilots

Within distance measurement communications, employing a singular pseudo-noise sequence for distance determination requires the sequence duration to be sufficiently extensive for unambiguous measurement outcomes. Nevertheless, conventional sequence capturing techniques demonstrate temporal requirements that correlate directly with sequence duration. Consequently, this creates challenges in rapidly obtaining extended-duration pseudo-noise sequences. In order to address this challenge, the current research employs numerous brief-duration distance measurement sequences as replacements for conventional reference signals. Furthermore, the concurrent communication capabilities of orthogonal frequency-division multiplexing frameworks are leveraged to integrate various distance measurement sub-sequences into unified composite sequences.

Orthogonal frequency-division multiplexing constitutes a multi-spectral concurrent communication architecture. According to Formula (5), it becomes evident that without inter-spectral interference effects, individual concurrent communication approaches may be regarded as autonomous waveform processing methodologies. Following reference sequence insertion in the spectral domain, for the kth reference element, based on Formula (1), we can observe that the waveform of the kth reference element in the temporal domain becomes(10)$$s_{p}^{k}(t)=p_{k}e^{j2\pi f_{k}t}$$
where h(0<h<N−1) represents the frequency carrier position where the reference element is located, and $P_{l}[h]$ denotes the lth reference symbol on the hth frequency carrier, l=0,1,2,…,∞.

From Formula (6), we can regard the kth reference element as an independent sub-waveform on the kth frequency carrier. Compared with the initial data stream, each sub-waveform’s data bit duration has been extended by N times. Thus, the data symbol with the initial duration of $T_{s}$ has been extended to $NT_{s}$. Since each sub-waveform is independent without inter-carrier interference, the temporal-domain continuous waveform in the corresponding frequency channel for the kth reference element can be equated as(11)$$s_{p}^{k}(t)=C_{k}(t)e^{j2\pi f_{k}t}$$
where $C_{k}(t)$ represents the distance measurement sub-sequence loaded on the h-th frequency carrier.

Under propagation medium influence, the detection unit strips the frequency carrier of the detected distance measurement waveform to obtain(12)$$r_{p}^{k}(t)=C_{k}(t-\tau +\hat{\tau })e^{j2\pi (f_{k}+f_{d}-{\hat{f}}_{d})(t-\tau +\hat{\tau })}$$
where $f_{d}$ represents the Doppler shift generated during propagation, ${\hat{f}}_{d}$ denotes the frequency deviation estimated by the detection unit, τ indicates the temporal delay generated by the waveform, and $\hat{\tau }$ represents the temporal delay compensated after synchronization at the detection unit. With perfect frequency deviation compensation and complete synchronization, i.e., when $f_{d}={\hat{f}}_{d}$, $\tau =\hat{\tau }$, Formula (12) can be expressed as(13)$$r_{p}^{k}(t)=C_{k}(t)e^{j2\pi f_{k}t}$$

Discretizing Formula (13) at the sampling frequency $f_{s}$, we can obtain the discrete form of the distance measurement reference element waveform:(14)$$r_{p}^{k}[n]=C_{k}[n-m]+w[n]$$
where m denotes the temporal phase sampling point of the distance measurement sequence corresponding to the temporal delay offset, and w[n] represents the discretization data corresponding to the noise. The correlation operation on Formula (14) yields(15)$$E_{k}(d)=\sum_{n=0}^{L_{k}-1}r_{p}^{k}[n]C_{k}^{*}[n-d]$$
where $C_{k}^{*}[n-d]$ represents the locally generated kth distance measurement sub-sequence, the temporal phase of which is shifted by d, and $L_{k}$ denotes the period of the kth distance measurement sub-sequence. A peak is obtained by adjusting d cyclically, at which point d represents the approximate temporal phase after the offset of the corresponding distance measurement sequence.

After coarse acquisition, the acquisition accuracy is only within ±0.5 code chips, and a more accurate code chip offset can be obtained through the tracking process. From the above analysis, each ranging code can be regarded as a single direct sequence spread spectrum signal without carrier interference. Consequently, this research employs an equivalent discretization monitoring loop founded on the direct spread baseband pseudo-noise delay-locked loop architecture. Relative to conventionally utilized monitoring loops, this approach eliminates the intricate implementation of the loop numerically controlled oscillator and the influence of oscillator initial phase uncertainty on pseudo-noise sequence monitoring precision.

The monitoring loop architecture is illustrated in Figure 7, which will monitor individual distance measurement reference elements concurrently. As demonstrated in Figure 7, each loop maintains an identical delay-locked loop structure, where k represents the sequential identifier of the distance measurement reference element. K denotes the quantity of reference element channels employed for distance determination. Additionally, $E_{P}$ constitutes the integration and accumulation outcome, $E_{E}$ represents the advanced correlation outcome, $E_{P}$ indicates the present correlation outcome, and $E_{L}$ signifies the delayed correlation outcome. Moreover, $C_{k}^{E}[n]$ and $C_{k}^{L}[n]$ employ pseudo-random sequences that advance or delay relative to the present branch pseudo-sequence by one discretization point, respectively. The $E_{E}$, $E_{P}$, and $E_{L}$ of the kth reference element can be mathematically expressed as(16)$$\left\{\begin{array}{l} \begin{array}{l} E_{E}=\sum_{n=0}^{L_{k}-1}r_{p}^{k}[n]C_{k}^{E*}[n]\\ E_{P}=\sum_{n=0}^{L_{k}-1}r_{p}^{k}[n]C_{k}^{P*}[n]\\ E_{L}=\sum_{n=0}^{L_{k}-1}r_{p}^{k}[n]C_{k}^{L*}[n] \end{array} \end{array}\right.$$
where $C_{k}^{P*}[n]$ represents the corresponding generated local pseudo-random sequence waveform following acquisition, and ϕ denotes the present temporal phase.

The phase detector of the loop adopts the normalized dot product power phase detection algorithm, and the output error of the phase detector is(17)$$e_{k}=\frac{E_{L}-E_{E}}{E_{P}}$$

ϕ will regulate the local regenerative pseudo-sequence generator to refresh a new set of pseudo-sequence temporal phase outputs, $C_{k}^{E}[n]$, $C_{k}^{P}[n]$, $C_{k}^{L}[n]$. The loop filtering mechanism can ensure smooth local pseudo-sequence temporal phase transitions. This research adopts an optimal second-order loop configuration, and the digital filtering function becomes(18)$$F(z)=K_{1}+\frac{K_{2}}{1-z^{-1}}$$
where $K_{1}$ and $K_{2}$ represent constant amplification coefficients.

Within this research’s computational analysis, to streamline the intricate implementation of conventional numerically controlled oscillators and baseband pseudo-sequence digital monitoring loops, the temporal phase shift direction of the pseudo-random sequence is regulated by the novel oscillator of the monitoring loop, with correlation spacing equivalent to two discretization point durations. The fundamental principle involves establishing an appropriate threshold Δ for comparing $e_{k}$ output. When $e_{k}$ output remains within [−Δ,Δ] boundaries, monitoring achieves lock status. Conversely, when $e_{k}$ output exceeds Δ or falls below −Δ, the pseudo-sequence temporal phase updates directionally leftward or rightward. The monitoring precision of this technique correlates with the system’s discretization frequency. Furthermore, the minimal sequence temporal phase offset achievable by the monitoring loop equals the offset of the sequence segment corresponding to one discretization point. Since the detection unit has acquired ±0.5 chips prior to monitoring, the monitoring procedure completes following a finite number of shift correlations.

### 3.2. Sidetone Ranging

Although using ranging pilots can achieve relatively large-range unambiguous distance measurements, their ranging precision is limited by code chip length and system sampling rate. To further improve ranging precision, this paper effectively inserts sidetone signals in the OFDM system for high-precision ranging. Sidetone ranging utilizes sinusoidal wave phase measurement principles to achieve distance measurement precision far superior to pilot ranging.

The sidetone working frequencies are selected based on traditional aerospace ranging principles: 300 kHz provides maximum unambiguous range with wavelength λ=1000 m, 1 MHz enables progressive ambiguity resolution, and 3 MHz achieves centimeter-level precision. This geometric progression follows classical multi-frequency sidetone configurations for ambiguity resolution. The OFDM integration maps these sidetone frequencies to specific subcarriers while preserving their physical ranging characteristics, representing an innovative fusion of aerospace sidetone technology with modern OFDM communications.

The 300 kHz, 1 MHz, and 3 MHz are sidetone signal working frequencies that maintain the physical principles of traditional aerospace sidetone ranging. Our innovation lies in carrying these sidetone signals as modulated information on OFDM subcarriers rather than direct transmission. The sidetone phase information $\phi =2\pi f_{tone}\times t_{delay}$ is modulated onto corresponding subcarriers and transmitted via the 34.8 MHz RF carrier, completely avoiding the large antenna requirements of traditional direct sidetone transmission while preserving centimeter-level ranging accuracy.

In OFDM systems, sidetone signals can be designed as specific frequency sinusoids, inserted into independent subcarriers outside of data and pilot signals. Let the sidetone signal be(19)$$s_{t}(t)=A_{t}e^{j2\pi f_{t}t}$$
where $A_{t}$ is the amplitude of the sidetone signal, and $f_{t}$ is the frequency of the sidetone signal. Similarly to pilot signals, sidetone signals at the receiving end after channel influence can be expressed as(20)$$r_{t}(t)=A_{t}e^{j2\pi (f_{t}+f_{d}-{\hat{f}}_{d})(t-\tau +\hat{\tau })}$$

In the case of complete frequency offset compensation, the received sidetone signal is simplified to(21)$$r_{t}(t)=A_{t}e^{j2\pi f_{t}(t-\Delta \tau )}$$
where $\Delta \tau =\tau -\hat{\tau }$ is the difference between the actual propagation delay and the compensated delay. By measuring the phase difference $\phi =2\pi f_{s}\Delta \tau +\phi_{offset}$ of the sidetone signal, distance information can be calculated:(22)$$\Delta R=\frac{c\cdot \phi_{t}}{4\pi f_{t}}$$

Since phase measurements can achieve very high precision, theoretically the precision of sidetone ranging can reach a small fraction of the carrier wavelength, far superior to code chip-level pilot ranging precision.

A key challenge of sidetone ranging is the distance ambiguity problem caused by phase periodicity. The unambiguous ranging range for a single frequency sidetone signal is(23)$$R_{\max}^{tone}=\frac{c}{2f_{t}}$$

This means that when the measured distance exceeds $R_{\max}^{tone}$, the phase will experience periodic repetition, leading to ambiguity in distance measurement.

In practical RF implementations, the fixed phase offset $\phi_{offset}$ can be eliminated through differential measurement between multiple sidetone frequencies or inter-satellite reference calibration techniques. This ensures that the integer ambiguity N in ranging calculations remains unaffected by synchronization imperfections, maintaining centimeter-level ranging accuracy in engineering deployments.

To solve the ambiguity problem of single-frequency sidetones, this paper designed a ranging system based on three different frequency sidetones. The system adopts high-, medium-, and low-frequency sidetone signals, respectively, 300 kHz (low-frequency sidetone $f_{L}$), 1 MHz (mid-frequency sidetone $f_{M}$), and 3 MHz (high-frequency sidetone $f_{H}$).

In this multi-frequency sidetone design, the maximum unambiguous ranging range is determined by the lowest frequency sidetone signal:(24)$$R_{\max}^{multi-tone}=\frac{c}{2f_{L}}=\frac{3\times 10^{8}}{2\times 300\times 10^{3}}=500$$

While ranging precision is mainly determined by the highest-frequency sidetone signal. For sidetones with frequency $f_{H}$, the distance measurement precision can be expressed as(25)$$\sigma_{R}=\frac{c}{4\pi f_{H}}\sigma_{\phi }$$
where $\sigma_{\phi }$ is the standard deviation error of phase measurement. Considering the influence of signal-to-noise ratio (SNR), phase measurement precision can be expressed as(26)$$\sigma_{\phi }\approx \frac{1}{\sqrt{2\cdot \mathrm{SNR}}}$$

### 3.3. Distance Measurement

This section details the distance measurement method of this system, which achieves large-range, high-precision distance measurement by combining the advantages of pilot ranging and sidetone ranging. Specifically, we first introduce the principle of pilot-based coarse ranging in Section 3.3.1, including how to use the Chinese Remainder Theorem to synthesize large-period composite codes to achieve large-range unambiguous ranging; then in Section 3.3.2 we discuss the precision ranging method based on sidetone signals, focusing on how to use multi-frequency sidetones to achieve high-precision ranging and how to use pilot ranging results to resolve the ambiguity problem for sidetone ranging.

#### 3.3.1. Pilot-Based Coarse Ranging

Following monitoring of individual distance measurement reference elements, the monitoring loop continuously outputs the temporal phase offset value of the corresponding sub-sequence. Subsequently, according to the Chinese Remainder Theorem, the detection unit can compute the total temporal phase offset and determine the communication distance.

The ambiguity resolution methodology based on the Chinese Remainder Theorem proceeds as follows.

Assuming that the reference element sub-sequences utilized for distance determination in this design are $p_{1}$, $p_{2}$, and $p_{3}$, and the periodic coprime of each distance measurement sub-sequence is $N_{1}$, $N_{2}$, and $N_{3}$, respectively, the maximum unambiguous range that can be determined by the distance measurement sequence is(27)$$(N_{1}N_{2}N_{3})T_{sym}c$$
where $T_{sym}$ is the duration of a single ranging code element, and c is the speed of light.

Let the corresponding integer digital segment offsets of the sequence monitoring loop output be $x_{1}$, $x_{2}$, and $x_{3}$, and determine $Y_{1}$, $Y_{2}$, and $Y_{3}$ that satisfy the following conditions:(28)$$\left\{\begin{array}{l} Y_{1}\mathrm{mod}N_{1}=1\\ Y_{1}\mathrm{mod}N_{2}=0\\ Y_{1}\mathrm{mod}N_{3}=0 \end{array}\right.,\;\left\{\begin{array}{l} Y_{2}\mathrm{mod}N_{1}=0\\ Y_{2}\mathrm{mod}N_{2}=1\\ Y_{2}\mathrm{mod}N_{3}=0 \end{array}\right.,\;\left\{\begin{array}{l} Y_{3}\mathrm{mod}N_{1}=0\\ Y_{3}\mathrm{mod}N_{2}=0\\ Y_{3}\mathrm{mod}N_{3}=1 \end{array}\right.$$

Subsequently the final accumulated integer offset Y based on the individual sub-sequence integer segment offsets becomes(29)$$Y=(x_{1}Y_{1}+x_{2}Y_{2}+x_{3}Y_{3})\mathrm{mod}(N_{1}N_{2}N_{3})$$

The determined communication distance is(30)$$(Y+Y_{i})T_{sym}c$$
where $Y_{i}$ represents the fractional sequence piece offset of the monitoring sequence loop output.

The unified sequence element displacement demonstrates sensitivity to individual element displacements monitored by respective distance measurement sequences. If monitoring of a single distance measurement sequence encounters inaccuracies, the comprehensive computational outcome experiences numerous calculation discrepancies. Nevertheless, employing several distance measurement sequences for distance determination ensures that fractional element displacements across various sequences remain uniform. Furthermore, simultaneous phase displacement generation by all distance measurement sequences results in equivalent phase displacement magnitude within the comprehensive unified sequence. For example, assuming that the integer digital segment offsets of the sequence monitoring loop output are $x_{1}+c$, $x_{2}+c$, $x_{3}+c$, the final offset $Y_{C}$ based on element offset monitored by each distance measurement sequence will be(31)$$Y_{c}=Y+Y_{i}+c$$

Consequently, leveraging this characteristic enables the detection unit to automatically rectify monitoring outcomes based on acquisition and detection results from individual distance measurement sequences.

It should be observed that pilot quantity varies with the subcarrier count employed in orthogonal frequency-division multiplexing systems. Subsequently, the quantity of distance measurement sequences utilized may fluctuate correspondingly. Within practical LEO satellite constellation operational scenarios, merely several brief-duration distance measurement sequences suffice to satisfy inter-satellite distance determination specifications.

Following the selection of necessary distance measurement sequences, these sequences can be repurposed as reference signals. Moreover, the quantity of distance measurement sequences correlates with the maximum unambiguous range of respective measurement sequences. Table 1 presents the maximum unambiguous range for varying quantities of periodic coprime distance measurement sequences operating at 0.6 MHz code frequency.

#### 3.3.2. Sidetone Ranging for Precise Measurement

Pilot ranging provides large-range but relatively low-precision distance estimation, while sidetone ranging provides high-precision but range-limited distance measurement. By combining these two measurement methods, large-range, high-precision distance measurement can be achieved.

The distance obtained from sidetone ranging can be expressed as(32)$$R_{tone}=\frac{c\cdot \phi_{t}}{4\pi f_{t}}+m\cdot \frac{c}{2f_{t}}$$
where $\phi_{t}$ is the measured phase difference, and m is an integer, representing distance ambiguity. The key issue is determining the correct ambiguity integer m.

Using the pilot ranging result as a reference, the ambiguity integer can be calculated:(33)$$m=\mathrm{round}\left(\frac{2f_{t}\cdot R_{pilot}}{c}-\frac{\phi_{t}}{2\pi }\right)$$
where $R_{pilot}$ is the distance value obtained from pilot ranging, and round represents taking the nearest integer. This method requires that the error of pilot ranging should be less than half of the sidetone ranging’s unambiguous range, i.e.,(34)$$\Delta R_{pilot}<\frac{c}{4f_{t}}$$

Once the correct ambiguity integer is determined, the final high-precision distance measurement value can be obtained by combining the pilot ranging and sidetone ranging results:(35)$$R=\alpha \cdot R_{pilot}+(1-\alpha )\cdot R_{tone}$$
where α is the weight coefficient, dynamically adjusted according to the reliability of the two measurement methods. In practical applications, α is usually close to 0 to fully utilize the high-precision characteristics of sidetone ranging.

Furthermore, to further improve ranging performance, the system can adaptively adjust sidetone frequencies and ranging code configurations according to communication environment and distance requirements. For example, in short-distance communications, the number of ranging pilots can be reduced and the number of sidetone signals increased to improve spectral efficiency; in long-distance communications, the number of ranging pilots is increased and appropriate sidetone frequencies are selected to ensure the unambiguity and reliability of ranging.

It should be noted that this theoretical ranging precision assumes ideal conditions including perfect clock synchronization, calibrated hardware phase offsets, and stable environmental conditions. Practical implementations require comprehensive error budgeting addressing: (1) clock drift and synchronization errors (typically ±10–100 ns), (2) hardware calibration uncertainties and temperature-induced phase variations, (3) relativistic effects in LEO orbits (~10 cm range bias), and (4) residual atmospheric delays.

Through the above methods, this system achieves high-precision, large-range, high-reliability distance measurement, providing precise positioning capability for OFDM communication systems and meeting the requirements of various application scenarios.

While the proposed method demonstrates centimeter-level ranging accuracy in simulations, practical implementation requires comprehensive error modeling similar to GNSS systems. Hardware biases (transmitter/receiver clock and phase offsets), atmospheric delays, and multipath effects will significantly impact sidetone phase measurements. The mature error compensation theories developed in GNSS systems provide valuable references for our future system implementation.

## 4. Simulation Results

In Section 2, we detailed the design of an integrated method for pilot ranging and sidetone ranging based on OFDM systems, proposing a new signal system; in Section 3, we conducted theoretical analysis of the ranging principles of this scheme. The simulation framework employs idealized conditions to validate the proposed algorithm’s intrinsic performance. Real-world deployment would require additional modeling of hardware non-linearities, environmental variations, and operational constraints. To verify the effectiveness of the proposed method, this section will comprehensively evaluate its performance through system simulation.

In the simulation design, we constructed a complete OFDM transmission and reception system, simulating the actual transmission environment including channel noise and interference. The transmitter inserts different ranging codes as pilots in the frequency domain, while also adding sidetone signals at specific subcarrier positions; the receiver implements channel estimation, coarse distance measurement, and high-precision ranging through corresponding processing. To ensure the reliability and reproducibility of the simulation results, Table 2 lists the key parameter settings used in this simulation.

For LEO inter-satellite link scenarios, the simulation adopts a free-space path loss model combined with AWGN channel characteristics. The path loss is calculated using the standard free-space propagation formula:(36)$$L_{fs}(dB)=20\log_{10}(4\pi d/\lambda )+L_{margin}$$
where d represents the inter-satellite distance (typically 1–15 km for LEO constellations), λ=c/f is the carrier wavelength (λ≈8.6 m for 34.8 MHz carrier), and $L_{margin}\approx 3\;\mathrm{dB}$ accounts for implementation margins and minor atmospheric effects in the space environment. For a typical 10 km inter-satellite distance at 34.8 MHz, the free-space path loss is approximately 88 dB.

The SNR values specified in Table 2 represent the post-compensation equivalent SNR after path loss correction, allowing direct evaluation of the proposed algorithm’s performance under realistic signal levels. This modeling approach is justified for LEO inter-satellite communications due to: (1) line-of-sight propagation with minimal multipath effects, (2) predictable orbital geometry enabling precise path loss prediction, (3) negligible atmospheric attenuation for space-based propagation paths, and (4) focus on intrinsic algorithm performance rather than link budget analysis. The simplified AWGN assumption effectively captures the dominant noise characteristics while enabling clear performance evaluation of the dual-mode ranging and channel estimation algorithms.

As demonstrated in Table 2, three distinct distance measurement sequences are chosen as reference elements in this computational analysis, with periodicities of 15, 19, and 23 chips, respectively. Based on Equation (14), the maximum unambiguous distance determination range that these three distance measurement sub-sequences can achieve is 3,277,500 m, which can be comprehensively applied to most LEO satellite constellation inter-satellite distance determination scenarios. Furthermore, this research developed a tri-frequency auxiliary tone distance determination framework, employing 300 kHz, 1 MHz, and 3 MHz auxiliary tone waveforms. This configuration, according to Equation (24), possesses a maximum unambiguous distance determination span of 500 m and theoretical accuracy achieving millimeter level.

To validate the effectiveness of the proposed interpolation method, we compare it with conventional approaches. Figure 8 shows the channel estimation mean squared error (MSE) performance comparison among linear interpolation, cubic spline interpolation, and our proposed dual-mode method.

As demonstrated in Figure 8, the proposed method achieves approximately 15–25% MSE reduction compared to cubic spline interpolation across different SNR conditions, while cubic spline outperforms basic linear interpolation as expected. The consistent performance improvement validates the effectiveness of integrating pilot and sidetone information in the channel estimation process.

Figure 9 illustrates the bit error rate performance comparison of medium response estimation utilizing conventional reference elements, distance measurement reference elements, and the integrated methodology of distance measurement reference elements with auxiliary tone waveforms.

As shown in Figure 9, the combined channel estimation scheme using composite ranging subcodes and sidetone signals performs best. The three methods show small differences in low-SNR regions, but when SNR > 2 dB, the composite scheme significantly outperforms traditional pilots. The combined scheme reduces BER to approximately 10^−3^ at 6 dB SNR, saving about 3 dB power under the same BER conditions compared to traditional schemes.

After channel estimation, the receiver continues using ranging codes and sidetone signals for ranging. Due to different working mechanisms, their combination provides more accurate distance measurement results. Figure 10 shows the acquisition probability of composite codes combined with sidetone signals and each ranging code under different SNRs. Code1, Code2, and Code3 correspond to the three ranging subcodes with periods of 15, 19, and 23 chips, which are used as pilots in the OFDM system and perform ranging functions. “Composite Code + Sidetone” in the legend represents the overall acquisition performance of the composite code combined with sidetone signals.

Figure 10 demonstrates that the detection unit can reliably capture individual distance measurement sequences when signal-to-noise ratio reaches −5 dB or higher. Within orthogonal frequency-division multiplexing implementation scenarios, system signal-to-noise ratio typically exceeds 10 dB; therefore this framework’s acquisition capability satisfies implementation requirements.

Following sequential capture of distance measurement sequences and auxiliary tone waveforms, the detection unit monitors the three distance measurement sequences concurrently. The monitoring loop architecture is presented in Figure 7. The average monitoring accuracy achieved by monitoring the three distance measurement reference elements and auxiliary tone waveforms under various signal-to-noise ratios is presented in Figure 11.

Figure 11 demonstrates that when signal-to-noise ratio exceeds 2 dB, the monitoring loop achieves a stable operational state, attaining high-accuracy sequence temporal phase monitoring. Moreover, monitoring accuracy correlates proportionally with pseudo-sequence periodicity. Consequently, the monitoring loop employed in this research performs exceptionally within the orthogonal frequency-division multiplexing framework. Based on monitoring loop output outcomes, we can determine unified sequence element displacement and accomplish coarse distance determination utilizing Equations (16) and (17).

Concurrently, the detection unit retrieves and processes the three auxiliary tone waveforms. Figure 12 and Figure 13 illustrate the temporal phase determination accuracy and distance determination accuracy of auxiliary tone waveforms under various signal-to-noise ratios.

Figure 12 shows the phase measurement precision of three frequency sidetone signals. The high-frequency sidetone has the largest phase error, while the low-frequency sidetone has the smallest, indicating stronger noise resistance in low-frequency sidetones. Errors change dramatically in low-SNR regions but tend to stabilize in high-SNR regions, with all sidetone phase errors below 2 degrees under 30 dB conditions.

Figure 13 shows that low-frequency sidetones have the largest distance measurement errors, while high-frequency sidetone signals have the smallest, contrary to phase error trends. This verifies Equation (25): high-frequency sidetones have shorter wavelengths, providing higher ranging precision. As SNR increases, ranging errors decrease significantly, with high-frequency sidetone precision reaching approximately 0.5 cm at 20 dB and all three-frequency sidetones achieving centimeter-level precision, far superior to traditional methods.

The comparison in Figure 14 shows that traditional pilot ranging has errors of about 100 m, almost unchanged with SNR; the composite ranging subcode method reduces errors to 10–30 m; while the proposed method performs best, with ranging errors significantly decreasing as SNR increases, from about 0.5 m at low SNR to about 0.02 m at high SNR, 3–4 orders of magnitude lower than traditional methods.

Figure 15 demonstrates the adaptive ranging mode’s switching mechanism and performance in dynamic ranging processes.

As shown in Figure 15, the entire system dynamically switches ranging modes based on an error threshold (50 m). The top subplot shows measured distance changes over time, indicating the measurement target moves within this distance range. The middle subplot displays real-time error values, and when measurement errors approach or exceed the threshold, the system automatically switches from simplified to complete ranging mode, as shown in the bottom subplot.

Figure 16 and Figure 17 show Doppler effects on the proposed system’s communication and ranging performance at different speeds. As shown in Figure 16, the proposed method maintains low error rates in the 0–100 m/s speed range, improving communication reliability by about 50% at high speeds of 100 m/s. Figure 17 compares three ranging methods’ performance in Doppler environments. The integrated ranging method performs best across the full speed range, maintaining approximately 43 m ranging precision even at 100 m/s, demonstrating its superiority in high-dynamic environments.

In summary, simulation results verify the effectiveness of the proposed method combining pilot ranging with sidetone ranging and the adaptive ranging mode. This system, within the OFDM framework, simultaneously achieves high-quality channel estimation and high-precision distance measurement, providing a feasible technical solution for integrated LEO satellite communication and ranging systems. The path loss modeling demonstrates that the proposed system maintains robust performance under realistic LEO inter-satellite propagation conditions, validating its practical applicability for constellation deployment.

## 5. Conclusions

This paper proposes an integrated OFDM signal system combining sidetone signals for communication and measurement, utilizing short-period pseudorandom codes as composite pilots for wide-range ranging, while inserting multi-frequency sidetone signals into subcarriers for precise ranging. The composite code resolves sidetone signal ambiguity problems, achieving centimeter-level precision. An adaptive ranging mode dynamically switches based on error thresholds, improving spectral efficiency while ensuring precision. Simulation results show the system achieves pilot acquisition when SNR > −5 dB, with high-frequency sidetone signals (3 MHz) reaching centimeter-level ranging precision; the dual-mode channel estimation algorithm saves about 3 dB power compared to traditional methods at the same bit error rate. The proposed dual-mode channel estimation demonstrates 15–25% MSE improvement over conventional cubic spline methods while maintaining significantly lower computational complexity.

The proposed scheme is suitable for OFDM LEO satellite communication systems and scenarios requiring high-precision ranging. However, some limitations remain in this research, with future research directions including:

1. Doppler frequency shift compensation techniques in high-speed mobile environments, especially for frequency offset-sensitive sidetone phase measurements.

2. Optimizing sidetone frequency configurations to balance ranging precision, unambiguous distance, and spectral efficiency in specific scenarios.

3. Comprehensive error modeling and compensation algorithms drawing from mature GNSS experience, including hardware bias calibration, atmospheric delay correction, and multipath mitigation for practical satellite systems.

4. Integer ambiguity resolution algorithms incorporating multiple error sources and real-time correction mechanisms.

5. Integration of PAPR reduction techniques to optimize power efficiency while maintaining ranging accuracy in practical satellite systems.

6. Extension to practical Ka/Ku band implementations with frequency-dependent parameter optimization, propagation modeling, and antenna design analysis for engineering deployment.

7. Hardware imperfection modeling and real-time calibration algorithms for practical satellite system deployment.

Through deeper research in these directions, the integrated signal system proposed in this paper is expected to play a greater role in future LEO satellite constellation systems and other communication applications requiring precise positioning.

## Figures and Tables

**Figure 1 sensors-25-05890-f001:**
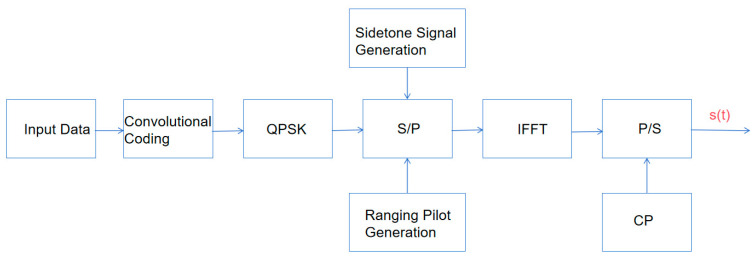
Block diagram of the OFDM transmitter.

**Figure 2 sensors-25-05890-f002:**
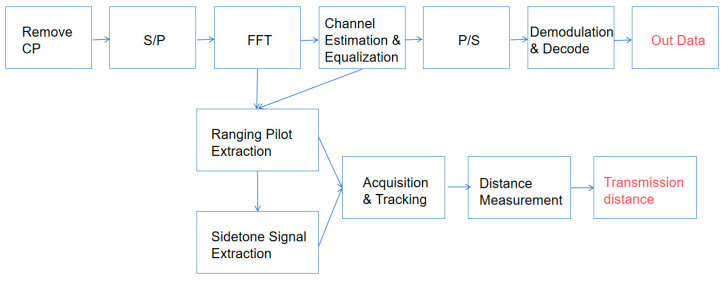
Block diagram of the OFDM receiver.

**Figure 3 sensors-25-05890-f003:**
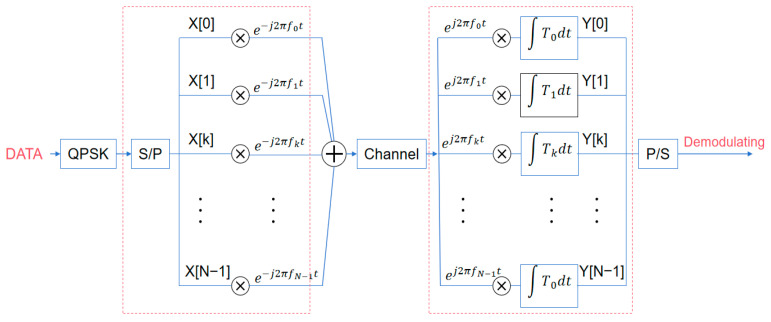
OFDM modulation and demodulation principle block diagram.

**Figure 4 sensors-25-05890-f004:**
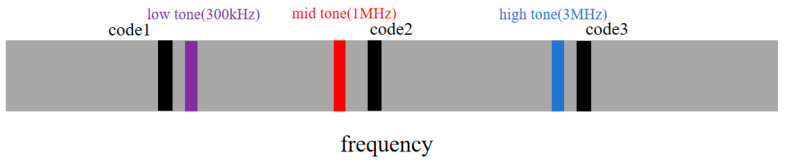
Frequency-domain pilot and sidetone signal insertion diagram (complete ranging mode).

**Figure 5 sensors-25-05890-f005:**
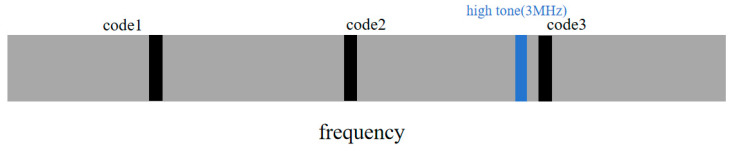
Frequency-domain pilot and sidetone signal insertion diagram (simplified ranging mode).

**Figure 6 sensors-25-05890-f006:**
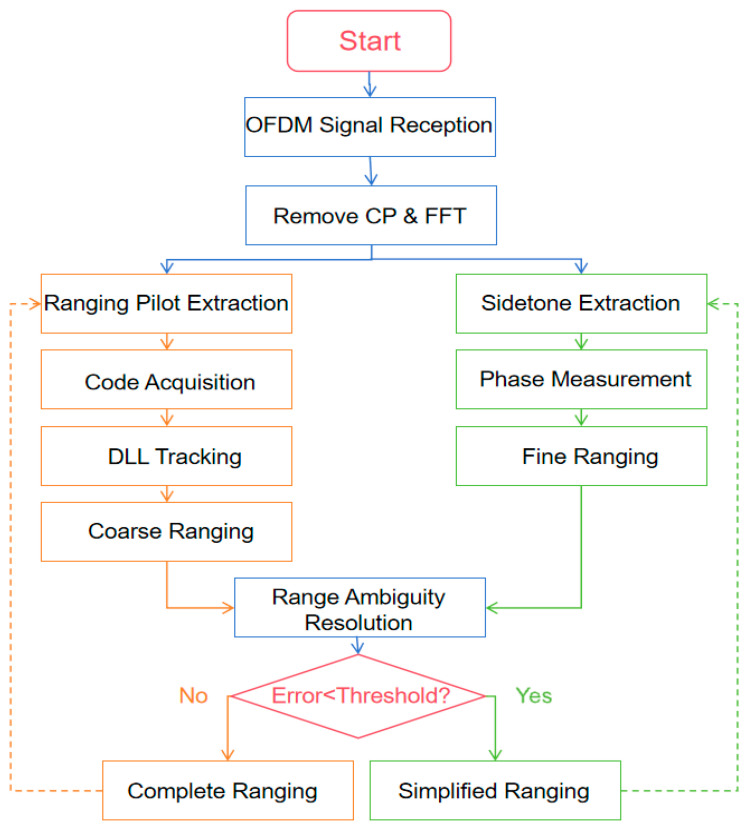
Overall algorithm flow diagram.

**Figure 7 sensors-25-05890-f007:**
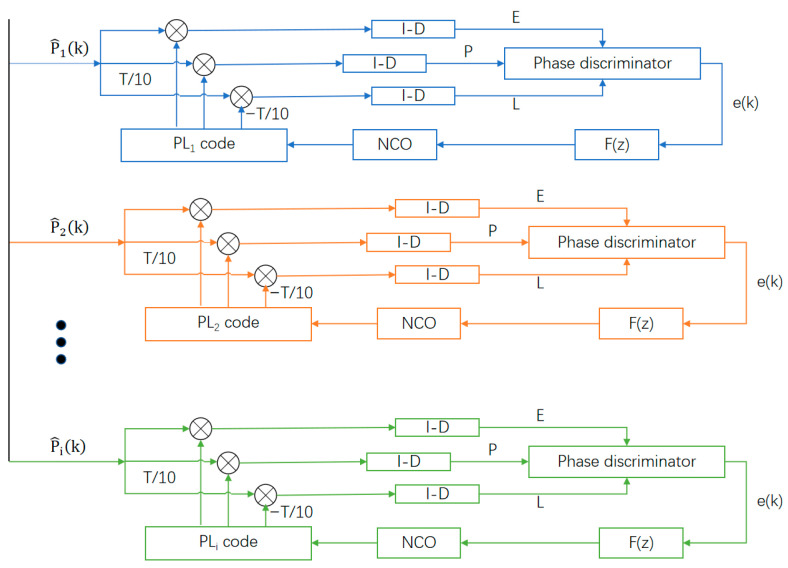
Pilot parallel tracking loop.

**Figure 8 sensors-25-05890-f008:**
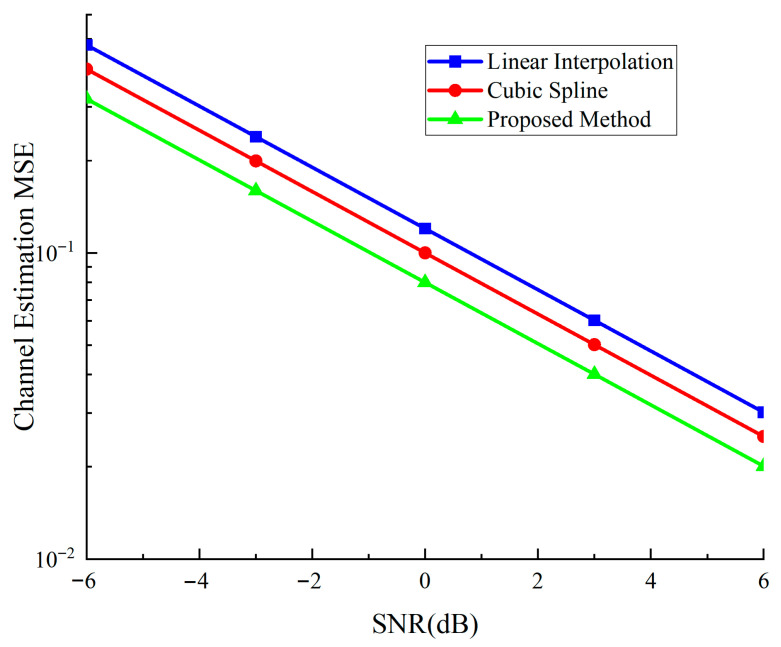
Channel estimation MSE performance comparison among different interpolation methods under AWGN channel conditions.

**Figure 9 sensors-25-05890-f009:**
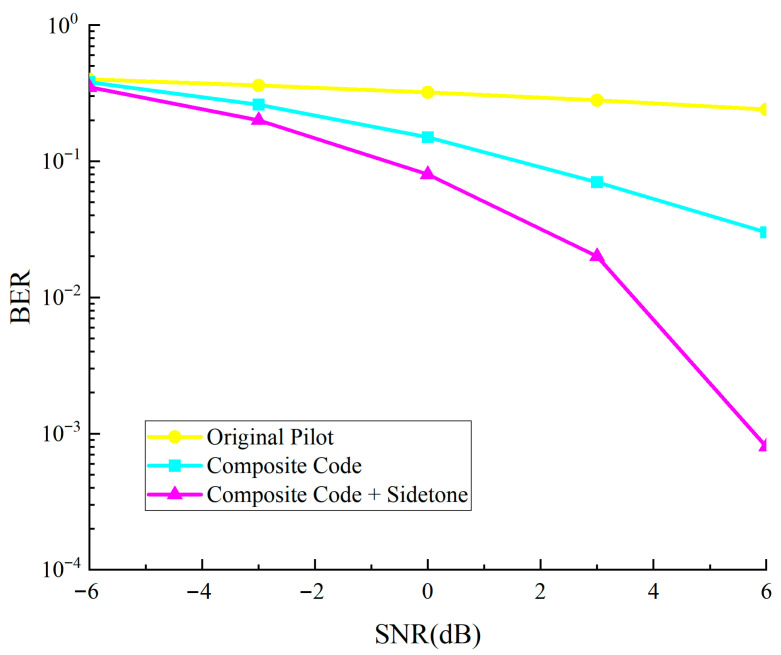
Performance comparison chart of different channel estimation methods.

**Figure 10 sensors-25-05890-f010:**
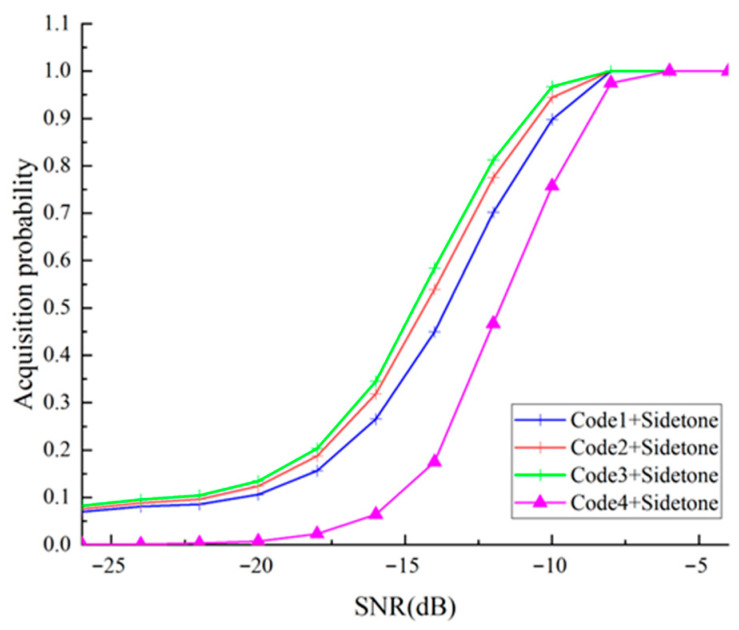
Acquisition probability of different ranging codes.

**Figure 11 sensors-25-05890-f011:**
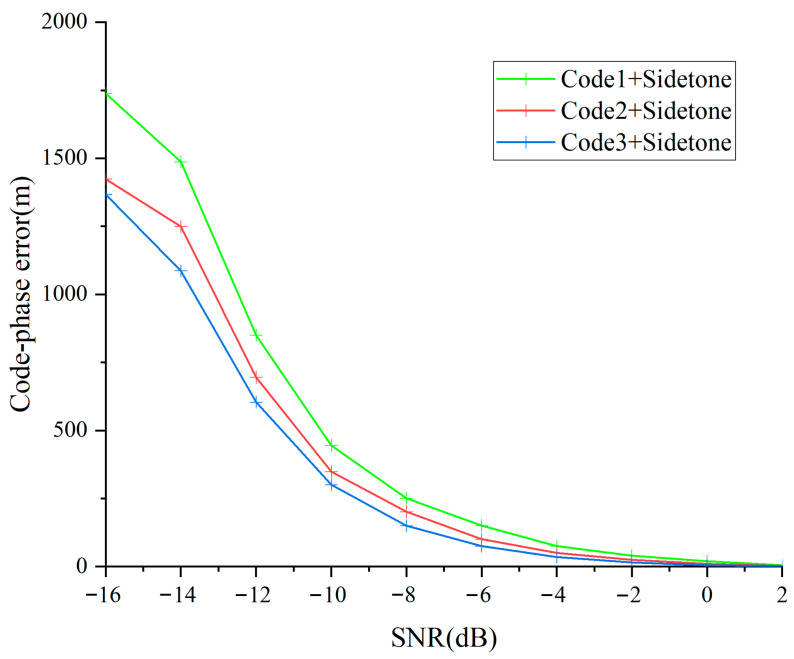
Tracking accuracy of different ranging codes.

**Figure 12 sensors-25-05890-f012:**
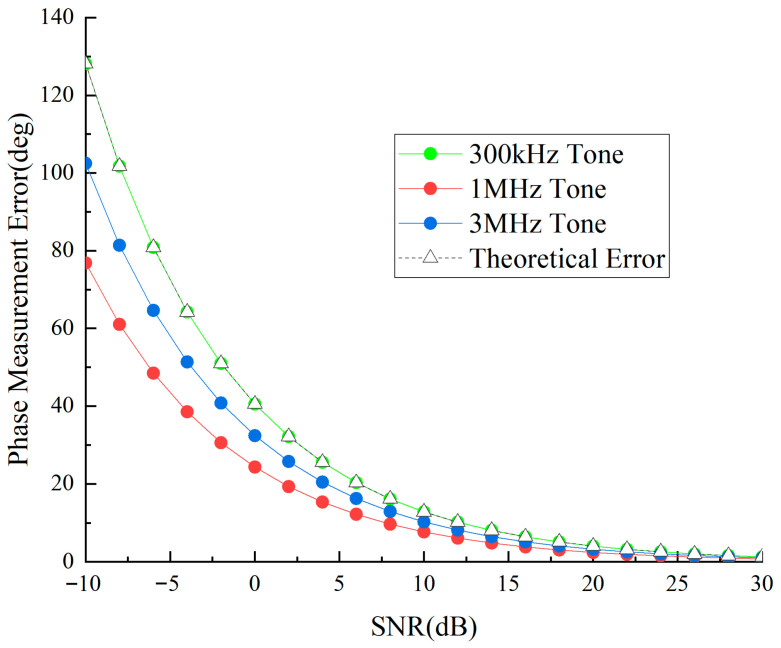
Sidetone phase measurement precision versus SNR curve.

**Figure 13 sensors-25-05890-f013:**
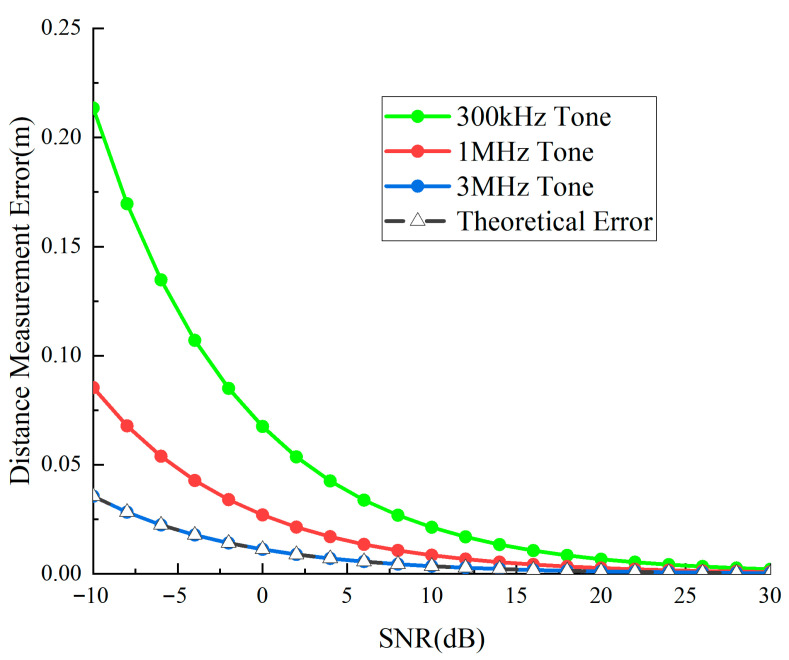
Sidetone distance measurement precision versus SNR curve.

**Figure 14 sensors-25-05890-f014:**
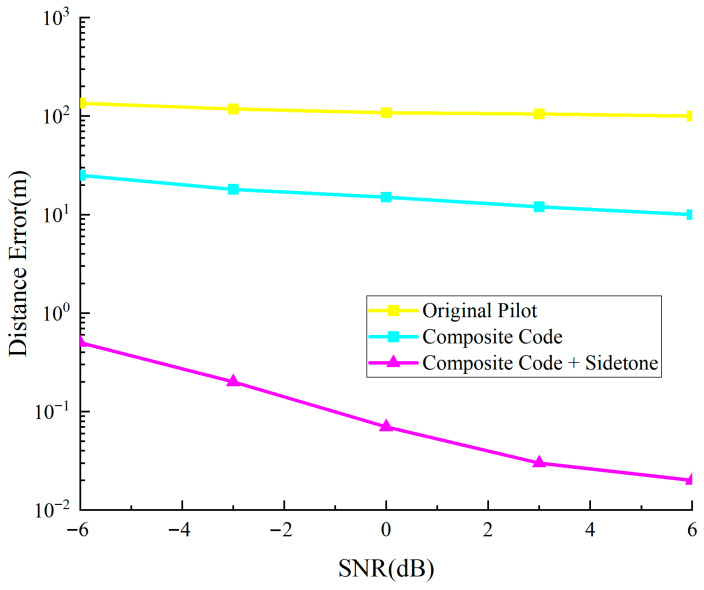
Precision comparison chart of original pilot and combined signal system ranging.

**Figure 15 sensors-25-05890-f015:**
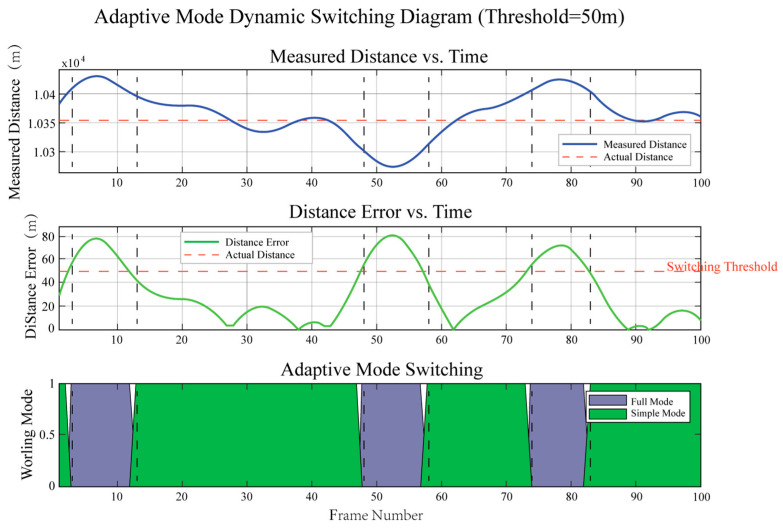
Performance analysis of adaptive ranging mode dynamic switching.

**Figure 16 sensors-25-05890-f016:**
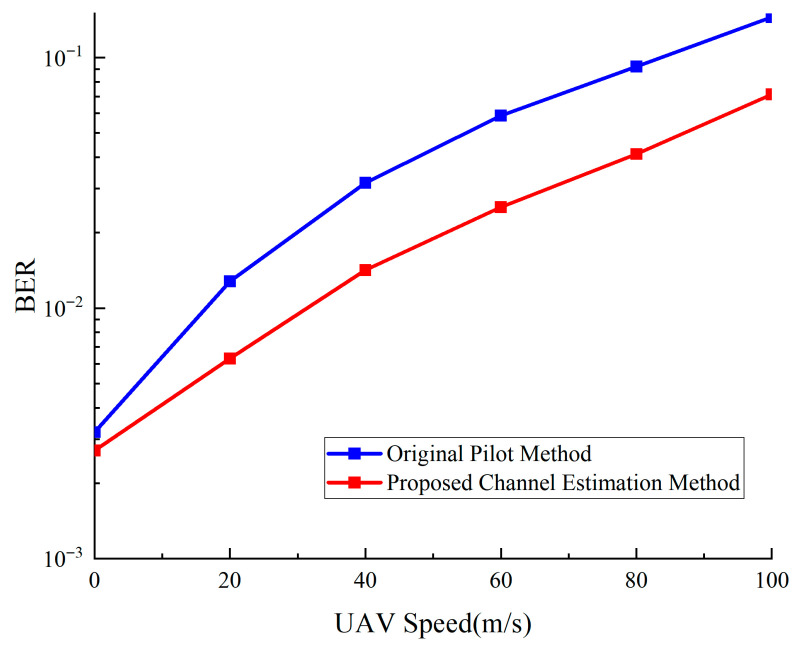
Effect of Doppler effect on communication performance.

**Figure 17 sensors-25-05890-f017:**
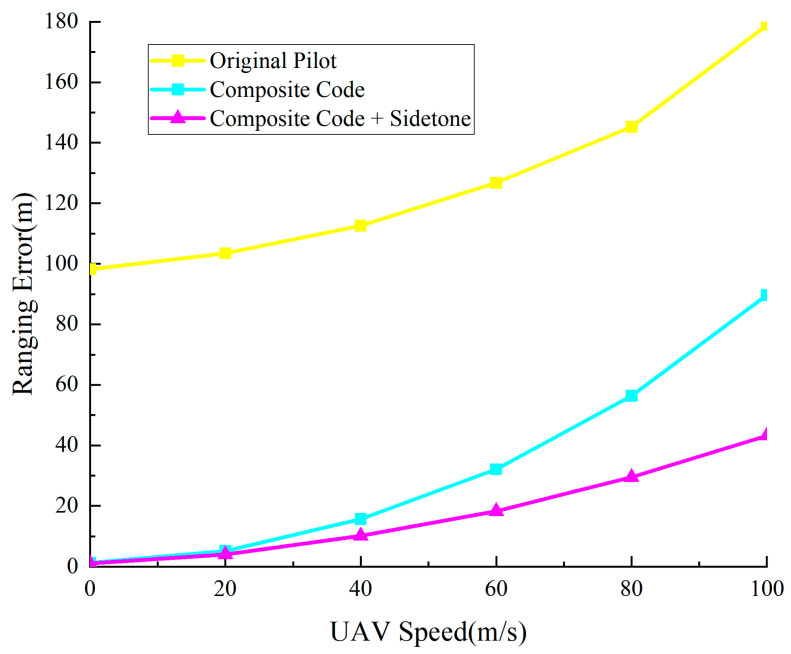
Effect of Doppler effect on ranging precision.

**Table 1 sensors-25-05890-t001:** Maximum unambiguous distance of coprime ranging codes with different periods.

Number	Period	Maximum Unambiguous Distance
2	2, 7	7 km
3	2, 7, 11	77 km
4	2, 7, 11, 15	1155 km
5	2, 7, 11, 15, 19	21,945 km
6	2, 7, 11, 15, 19, 23	504,735 km

**Table 2 sensors-25-05890-t002:** Simulation parameters.

Parameter	Value
Number of total subcarriers	N=64
Length of cyclic prefix	$N_{cp}=16$
Modulation scheme	QPSK
Signal frequency f	34.8 MHz (algorithm validation frequency; extensible to Ka/Ku bands)
$\mathrm{Sampling}\;\mathrm{frequency}\;f_{s}$	348 MHz
$\mathrm{Subcarrier}\;\mathrm{frequency}\;f_{sc}$	0.6 MHz
$\mathrm{Code}\;\mathrm{frequency}\;f_{\;\mathrm{code}\;}$	0.6 MHz
$\mathrm{Code}\;\mathrm{sampling}\;\mathrm{frequency}\;f_{s-\mathrm{code}\;}$	6 MHz
Channel mode	AWGN with free-space path loss
DLL bandwidth	1 Hz
DLL correlator spacing	0.25 chip
Number of pseudo-random codes	3
The period of pseudo-random codes	15, 19, 23
The period of pseudo-random codes	300 kHz, 1 MHz, 3 MHz
Path loss model	$L_{fs}=20\log_{10}(4\pi d/\lambda )$
SNR definition	Post-compensation equivalent SNR

## Data Availability

The data presented in this study were generated through MATLAB R2022b simulations based on the parameters and methodology described in Section 4. All simulation parameters are provided in Table 2 of this manuscript. The simulation code and generated datasets supporting the reported results are available from the corresponding author upon reasonable request. No publicly archived datasets were analyzed in this study, as all data were generated specifically for this research through computational simulations of the proposed OFDM-based integrated ranging and communication system.

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
