# Peer review of "Integrated Communication and Navigation Measurement Signal Design for LEO Satellites with Side-Tone Modulation"

_sensors, 2025, doi:10.3390/s25185890_

Round 1
Reviewer 1 Report
Comments and Suggestions for Authors
Authors designed an OFDM signal for LEO satellites capable of high-precision ranging. Although claiming exceptionally high accuracy, authors appears to have limited understanding of the critical issues in existing GNSS ranging technologies and failed to account for key engineering practical constraints.
- All the experimental result figures (Fig.8- Fig.16) in the paper are bitmaps. It is better to use clearer vector graphics.
- The cross-references for formulas, figures, paragraphs, and references are all invalid. Even if written in Word, cross-references can be retained when converted to PDF. Please add cross-references to facilitate reading.
- Some formulas (Eq.26) contain Chinese characters.
- Why separate the frequencies of ranging pilots and sidetone signals? After demodulation, ranging pilots also have phase observations that can be used, which would also save the trouble of weighting.
- The author's use of ranging pilots and sidetone signals for coarse ranging and precise measurement in this paper is basically consistent with the logic of using ranging codes for pseudorange measurement and carrier phases for precise positioning in existing GNSS systems. However, the author seems to have limited understanding of key issues in existing satellite navigation technologies. For example, in precise point positioning using carrier phases, tropospheric delay, frequency-dependent ionospheric delay, and hardware bias delay in signal propagation will significantly affect the resolution of integer ambiguity. The impact of these key points is not mentioned in this paper.
- The author designed sidetone signals of 300kHz, 1MHz, and 3MHz. Mathematically, using a very low-frequency signal like 300kHz is naturally beneficial for solving integer ambiguity, but it is fundamentally impractical in engineering. The wavelength of a 300kHz signal is 1000 meters, which would require extremely large antennas for both transmitting and receiving signals. Essentially, 300K-3MHz is not suitable for satellite communication.
Author Response
Dear Editor,
We would like to express our sincere gratitude for taking the time to review our manuscript "Integrated Communication and Measurement Signal Design for LEO Satellites With Side-Tone Modulation" and for providing us with the opportunity to revise our work. We deeply appreciate your valuable suggestions and comments, which we have carefully considered and addressed in detail.
Our point-by-point responses to all comments are provided in the attached document "Reply", and the revised manuscript with all changes clearly marked has been uploaded as "revised manuscript" for your review.
We sincerely appreciate the valuable guidance provided by the editor and reviewers. Each revision has contributed to the improvement and refinement of our manuscript. We hope that the revised manuscript now meets the publication standards. Should there be any areas requiring further modification or if you have additional suggestions, we would be grateful for your continued guidance.
We wish you good health, success in your work, and happiness in life.
Sincerely yours, Xue Li August 16, 2025 Ctr Commun and Tracking Telemetry Command, Chongqing University, Chongqing 400044, China

Reviewer 2 Report
Comments and Suggestions for Authors
The paper uses an OFDM signal that integrates short period coprime ranging codes together with sidetone tones then fuses code based and tone based channel estimates through a distance weighted rule for the problem of interference from separated communication and measurement functions in LEO satellites. The paper can be improved. Please see comments below:
- A problem with OFDM is the high peak-to-power average ratio (PAPR). It can be good to mention methods that generate waveforms with adjustable PAPR such as [R1]. Kindly include.
- Please explain why you choose the three sidetone frequencies at 300 kHz, 1 MHz, and 3 MHz.
- In equation (8), the authors use a linear weight that depends on subcarrier distance alone, but it can be good to have an adaptive weight that also scales with instantaneous SNR so that noisy sidetone bins contribute less even when spectrally close.
- The authors mention “improved spline interpolation algorithm” but dont mention in which sense. Please in simulations compare with the “ spline interpolation algorithm” alone. Also, compare with standard cubic spline so that the gain in mean squared error becomes clear. Also, besides spline interpolation, there are methods for spatio-frequential smoothing that can be included.
- Compare the least square approach given in equation (6) and (7) with that if minimum mean squared error equalization approach instead.
- Some formatting issues should be fixed like $f_{s- code }$ in Table 2 should be $f_{s-code }$. Also “10e-3 “ in the abstract should be “10^{-3}”.
- What is the path loss model assumed in simulations ?
- The figure resolution should be significantly enhanced such as the title of fig. 12 which is blurry.
- The paper contains typos like “the paper demonstrate” which should be “the paper demonstrates”. Also please avoid using congratulatory statements like “innovatively”. Also, the unit should be separated from the number such as “6dB” should be “6 dB”. Kindly revise.
References
[R1] “On Integrated Sensing and Communication Waveforms With Tunable PAPR,” in IEEE Transactions on Wireless Communications, vol. 22, no. 11, pp. 7345-7360, Nov. 2023, doi: 10.1109/TWC.2023.3250263
Author Response

(The authors gave the same response as above.)

Round 2
Reviewer 1 Report
Comments and Suggestions for Authors
- Line 419 indicates the use of a 34.8 MHz carrier frequency. Why was this frequency band chosen as the carrier? In practice, such a frequency is still too low for satellite communications due to constraints on antenna size and the longer wavelength, which is more prone to ionospheric reflection. More commonly used frequency bands such as Ka or Ku should be considered instead.
- Details regarding the selection of the carrier frequency should be introduced earlier, around line 184, where the signal format is first discussed.
- During carrier phase measurement, it is generally inappropriate to use the intermediate frequency (IF) representation as shown in Equations (19)–(21). Instead, the complete radio frequency (RF) signal including the carrier should be considered. During signal reception, since the local receiver clock cannot be perfectly synchronized with the satellite clock, a fixed phase offset is introduced even when the frequencies match (as illustrated in Figure 3: modulation and demodulation cannot both use the same carrier phase e^(j*2*pi*fi*t), one of them have to be e^(j*2*pi*fi*t + phi), resulting in a phase difference phi between them). This implies that the phase difference in Equation (22) includes not only the delay-induced phase shift but also this fixed offset. Such a fixed phase deviation may significantly impact integer ambiguity resolution.
- The overall assumptions in the article are overly idealistic. Many practical challenges in high-precision ranging have not been adequately addressed. Those issues should be mentioned in further work.
Reviewer 2 Report
Comments and Suggestions for Authors
The reviewer thanks the authors for addressing the comments.
Author Response
Dear Editor and Reviewers:
We sincerely thank you for your thorough review and positive evaluation of our manuscript "Integrated Communication and Measurement Signal Design for LEO Satellites With Side-Tone Modulation." We are grateful for your recognition that our research design is appropriate, methods are adequately described, results are clearly presented, and conclusions are well-supported by the findings. Your assessment that all figures and tables are clear and well-presented is particularly encouraging.
We appreciate your acknowledgment that the English language quality is satisfactory and does not require improvement. Your confirmation that we have adequately addressed the previous comments provides valuable validation of our revision efforts. The positive evaluation across all assessment criteria demonstrates the thoroughness of your review process and gives us confidence in the manuscript's readiness for publication.
We thank you for your time and expertise in reviewing our work. Your professional evaluation contributes significantly to the scientific peer review process and helps ensure the quality of published research in this field.
We wish you good health, success in your work, and happiness in life.
Sincerely yours,
Xue Li
September 3, 2025
Ctr Commun and Tracking Telemetry Command, Chongqing University, Chongqing 400044, China
